# Life Events in the Etiopathogenesis and Maintenance of Restrictive Eating Disorders in Adolescence

**DOI:** 10.3390/children10020376

**Published:** 2023-02-14

**Authors:** Giorgia Baradel, Diletta Cristina Pratile, Marika Orlandi, Arianna Vecchio, Erica Casini, Valentina De Giorgis, Renato Borgatti, Martina Maria Mensi

**Affiliations:** 1Department of Brain and Behavioral Sciences, University of Pavia, 27100 Pavia, Italy; 2Child Neurology and Psychiatry Unit, IRCCS Mondino Foundation, 27100 Pavia, Italy

**Keywords:** adolescents, anorexia nervosa, mental disorders, restrictive eating disorders, life events, preventions

## Abstract

Life events (traumatic and protective) may be critical factors associated with eating disorders and their severity. To date, there is little literature concerning the role of life events in adolescence. The main goal of this study was to explore in a sample of adolescent patients with restrictive eating disorders (REDs) the presence of life events in the year before enrolment and to characterize them according to timing. Furthermore, we investigated correlations between REDs severity and the presence of life events. In total, 33 adolescents completed the EDI-3 questionnaire to assess RED severity using EDRC (Eating Disorder Risk Composite), GPMC (General Psychological Maladjustment Composite), and the Coddington Life Events Scales—Adolescent (CLES-A) questionnaires to define the presence of life events in the last year. Of these, 87.88% reported a life event in the past year. A significant association emerged between elevated clinical GPMC and the presence of traumatic events: patients who had experienced at least one traumatic life event in the year before enrolment presented higher clinically elevated GPMC compared to patients who had not. These results suggest that obtaining early information about traumatic events in clinical practice may help prevent the occurrence of new events and improve patient outcomes.

## 1. Introduction

Restrictive eating disorders (REDs) are a heterogeneous group of eating disorders (EDs) which include anorexia nervosa (AN), atypical anorexia (A-AN), avoidant/restrictive food intake disorder (ARFID), and other specified EDs with restrictive characteristics [1]. Typical anorexia nervosa, the most studied and best known condition, is an eating disorder characterized by food restriction, fear of gaining weight, and altered perception of one’s body weight [1]. In comparison with other psychiatric disorders, eating disorders, and anorexia nervosa in particular, are characterized by a special complexity.

REDs show an increased incidence among young people in recent years, with 40–100 cases per 100,000 in females and 1–4 cases per 100,000 in males in the age group between 15 and 19 years and 13.4% of females and 7.1% of males affected in the age group between 9 and 14 years of age [2].

Moreover, REDs are linked with the highest mortality risk rate among psychiatric diseases, and their severity is associated with moderate to high levels of psychosocial and work impairment [3].

Multiple interactions between several risk factors, such as environment and individual biological vulnerability, are thought to be involved in the development of EDs [4]. It is worth mentioning that the recent SARS-CoV-2 pandemic had an effect on the development and exacerbation of eating disorders, in particular typical and atypical anorexia nervosa. This effect has been outlined by several studies; moreover, an American study reported that, among a sample of patients aged 8–18 years who were hospitalized for the first time during the lockdown, 33.3% reported that the only environmental factor correlated with the onset of typical or atypical anorexia nervosa was the lockdown itself [5].

Traumatic life events (LEs) have often been recognized as risk factors in the development of EDs and as strong determinants of the severity of the patient’s clinical presentation and psychiatric comorbidities [6,7]. Several studies have shown the significance of physical, sexual, and psychological abuse, emotional and physical neglect in EDs. Childhood sexual abuse is the most extensively studied form of traumatic experience related to EDs, and the literature data reveal that it may have role in the development of AN. However, there is no single interpretation regarding the correlation between the type of traumatic event and the type of ED: indeed, the difference between sexual and non-sexual early stressful life events in predisposing to a later ED is particularly controversial [8,9,10]. In addition, a family history of psychiatric pathology acts as a traumatic event in the development of AN, probably due to the family environment in which affectivity is negatively experienced and family cohesion is not strong [11]. Traumatic events can be either single or multiple and can be invasive and/or interpersonal: in the literature, it appears that severity of the clinical picture and onset of typical anorexia nervosa are not correlated with the number of traumatic events experienced or with their nature [4].

Furthermore, according to some authors [12], patients suffering from AN would refer to an internal ‘voice’ of the disorder, probably resulting from early life traumas and relational dynamics. It carries with it very strict behavioral rules; it seems to provide comfort, predictability and reassurance in adverse experiences, but it also proves hostile and belittling in the face of rule-breaking, revealing similar characteristics to abusers and bullies in childhood [13]. Some cognitive behavioral models suggest that early emotional abuse can lead to the development of negative core beliefs about oneself, which increases vulnerability to psychopathological disorders [12,14,15] and perpetuate AN by mimicking previous abusive relational dynamics [13]. Traumatic childhood experiences appear to be involved in biological changes in the stress response system. Some studies have found that adolescents with a history of childhood trauma had lower levels of salivary cortisol on awakening than the control group (affected patients without a history of trauma) [16]. Such hypoactivity could represent a vulnerability factor that compromises the ability of individuals to cope with stressful events, such as REDs [16].

A major review in 2017 clarified the state of the art regarding the role played by protective life events in the development of REDs. Protective life events could be more generic, such as family support and cohesion, or more specific, such as the importance given in the family to eating meals together, the avoidance of negative comments about weight, or growing up in a family where thinness or physical attractiveness are not overemphasized [17]. Personal protective factors also include the achievement of school goals, the construction of goals for the future and the aim to pursue their attainment, the ability to fit into society and play different and diverse roles, and problem solving. In addition, the social environment the patients relate with and are influenced by should be considered. Belonging to a cultural society that accepts a range of sizes and body shapes, playing a sport that emphasizes teamwork, and rejecting the exaltation of thinness have been noted to be social protective factors [18]. Finally, building a good relationship with food can be considered a protective factor: planning meals at certain times; eating a variety of foods from all food groups; introducing the right amount of food to maintain health and growth of the individual; eating flexibly, spontaneously, and for pleasure; and listening to what one’s body requires can induce a lower risk for developing an ED [18].

Considering the above, only a little attention has been paid so far to life events in adolescent patients with REDs [19]. Adolescence is a complex developmental period, and it has been classically considered as the predominant onset period of anorexia (14–18 years), although an earlier age of onset has been observed in recent years, with restrictive symptoms appearing as early as in prepubertal girls attending primary school [2].

Furthermore, high personal and health costs are associated with children and adolescent who have experienced trauma and they are at high risk of developing post-traumatic stress disorder (PTSD) and other negative emotional, behavioral, and mental health outcomes.

The present study aimed to bridge this gap in the literature with a three-fold aim: (a) to explore the presence of life events considered determinants for the development of physical or emotional disorders in the year before enrolment in the study population diagnosed with RED; (b) to explore the relationship between the presence of life events and the severity of RED and psychological maladjustment; and (c) to explore the relationship between the presence of traumatic and protective life events and the severity of RED and psychological maladjustment.

## 2. Materials and Methods

### 2.1. Participants

From March 2020 to May 2022, we enrolled 33 female adolescent patients (12 to 18 years old) who were referred to the Child Neurology and Psychiatry Unit of IRCCS Mondino Foundation in Pavia as inpatients, outpatients, or in day-hospital-regimen patients (Figure 1). Inclusion criterion for the study was a diagnosis of RED according to the Diagnostic and Statistical Manual of Mental Disorder (DSM-5) criteria [1]. RED diagnosis could include any restrictive and binge-eating/purging subtypes of AN, A-AN, ARFID or other specified EDs with restrictive characteristics. RED diagnoses were confirmed using the DSM-based Kiddie Schedule for Affective Disorders and Schizophrenia (K-SADS) [20]. Exclusion criteria were an insufficient understanding of the Italian language and belonging to a single-parent family. We also excluded patients with any comorbid neurological disorders (e.g., autism spectrum disorder or other psychiatric diseases), patients who presented intellectual disability as assessed using age-appropriate Wechsler intelligence scales, and personality disorders evaluated through the structured clinical interview for DSM-5 Personality Disorders (SCID-5 PD) [21].

This cross-sectional study was approved by the Ethics Committee of the Policlinico San Matteo in Pavia (Protocol ID: P-20170016006) and performed according to the Reporting of studies Conducted using Observational Routinely-collected health Data (RECORD) statement (Appendix A). All patients and their caregivers provided written informed consent to the study. The authors assert that all procedures contributing to this work comply with the ethical standards of the relevant national and institutional committees on human experimentation and with the Helsinki Declaration (1964) and its later amendments. The dataset is available upon request in Zenodo [22].

### 2.2. Procedures and Measures

Patients were interviewed about family and medical history and a child neuropsychiatrist carried out a psychiatric evaluation and a neurologic examination. Adolescents were asked to complete the Eating Disorders Inventory—3 (EDI-3) [23], a 91-item self-report questionnaire that evaluates psychological constructs known to be clinically relevant to REDs. The items, which are rated on a 6-point Likert scale (range: 1, never; 6, always), are grouped into 12 subscales: 3 of them specific to eating disorders and 9 focused on patients’ psychological characteristics. For the aims of the present study, two composite scores were considered to evaluate the severity of RED. The General Psychological Maladjustment Composite (GPMC) includes reports for low self-esteem (LSE), personal alienation (PA), interpersonal insecurity (II), interpersonal alienation (IA), interceptive deficits (ID), emotional dysregulation (ED), perfectionism (P), asceticism (A), maturity fear (MF). The Eating Disorder Risk Composite (EDRC) included reports of drive for thinness (DT), body dissatisfaction (BD), and bulimia (B). The outcome is non-clinical when the result obtained in each composite scale gives a score below the 70th percentile, clinical when the score is between the 70th and 85th percentile, and highly clinical when it is above the 85th percentile.

In addition, adolescent patients were asked to fill in the Coddington Life Events Scales—Adolescent (CLES-A) [24], a 50-item self-report form about life events that adolescents have experienced in the 12 months preceding enrollment. Life events could be either positive or negative and each with a specific value: the most recent events had a greater impact than the oldest ones. The higher the associated scores, the more stressful the life event was and required more psychological readjustment to overcome. However, it should be noted that the reaction to the event can be influenced by patient’s perception about positive or negative event.

The corresponding scores in Life Change Unit (LCU) were calculated in period 0–3 months (LCU-A), 0–6 months (LCU-B), 0–9 months (LCU-C), and 0–12 months (LCU-D) before enrollment. Each of these four periods has a specific cut-off. The score awarded is full when the event occurred in the previous 3 months, ¾ of the total when it occurred 4–6 months earlier, 1/2 if it occurred in the previous 7–9 months, and, finally, ¼ if it occurred between the previous 10 and 12 months. Scores resulted positive, and patients were considered at risk of developing physical or emotional disorders (supra-threshold CLES) if they exceeded the LCU-appropriate and the age-appropriate 75th percentile.

### 2.3. Statistical Analyses

The analyses were conducted using IBM SPSS 27 [25] and R [26] for Windows, setting *p* < 0.05. The chi-squared test χ^2^ was used to investigate the linear relationship between CLES-A and both EDI-3 EDRC and EDI-3 GPMC. Chi-squared test was also used to assess the relationship between the age of onset of the REDs and life events. Spearman correlation was used to verify the hypotheses that relate EDI-3 and the presence of traumatic or protective events because these variables are not parametric.

## 3. Results

In this study, 29 patients had a history of at least one traumatic event and as many patients with a history of at least one protective event, as Figure 2 shows. Regarding traumatic life events, 25 patients reported it to be family-related, 9 reported it to be social-relational, and 12 reported it to be personal. Notably, four patients (12.12%) described traumatic events in all three categories.

On the other hand, 17 patients reported protective life events of a familial nature, 20 patients reported social-relational events, and 15 patients reported personal events.

Thirty-two patients (96.97%) reported a history of life events during the 12 months before enrollment, either traumatic or protective. Eleven patients (33.33%) resulted positive for the CLES-A test, thus obtaining a score above the age cut-off. Of these patients, nine (27.27%) indicated a life event in the 3 months before enrolment.

It is possible to observe the trend of patients’ scores at CLES during the four periods taken into consideration: patients who showed supra-threshold scores 10–12 months before enrolment maintained supra-threshold scores in all three following periods. Moreover, 10 out of 11 patients with a positive test had experienced at least one event impacting enough to give an immediate supra-threshold score (30.30% of the total sample, 90.91% of the population who experienced supra-threshold life events).

There was no statistically significant result in the correlation between LCU calculated in all four periods examined and both EDRC (Figure 3) and GPMC (Figure 4).

A significant correlation was obtained between total traumatic events and EDI-3 GPMC: the more traumatic events the patient experienced, the greater the level of psychological maladjustment (Spearman’s Rho (32) = 0.346; *p* = 0.05). In contrast, the correlation between EDI-3 EDRC and traumatic events was not statistically significant (Spearman’s Rho (32) = 0.246; *p* = 0.17). No statistically significant correlations were found between the presence of protective life events and the severity of REDs. (EDI-3 GPMC: Spearman’s Rho (32) = 0.185; *p* = 0.30; EDI-3 EDRC Spearman’s Rho (32) = 0.216; *p* = 0.23). Table 1 shows the frequencies of the presence of traumatic events and EDI-3 GMPC severity in the sample.

## 4. Discussion

The primary aim of this study was to assess the presence of supra-threshold traumatic or protective life events in adolescent patients with restrictive eating disorders in the year before enrolment. In the study, 90.91% of the life events reported by the patients were related to the period furthest from enrolment and were followed by other life events more closely related to the access to our institute. This is in line with recent literature that clarifies how having experienced a significant life event makes the patient more susceptible to other subsequent events. Our findings are consistent with these theories; indeed, 29 out of 33 patients in our sample reported a supra-threshold life event in the year before enrollment. This is documented especially for those of traumatic nature [16,27].

Concerning the nature of life events, these are mainly family-related traumatic events, such as domestic violence, and physical, sexual, or emotional abuse. This is in line with the existing literature that has mainly focused on events occurring during childhood or early adolescence [12,14,28]. In this regard, a recent study provides data on the correlation between traumatic life events and restrictive eating disorders: 67% of the study population reported at least one traumatic event in adolescence (bullying, major loss, accident) and 19% associated the traumatic event with the sexual sphere. Of these, 57% reported having experienced a sexual assault and 30% a rape [14]. In addition, we have to consider that 51.52% of our patients have a family history of psychiatric pathology, with 24.24% of these concerning depression. Although these are preliminary results, they support the literature confirming that a positive family history of psychiatric pathology induces an increased susceptibility to the development of REDs and a worse clinical outcome [29].

On the other hand, regarding protective life events, 60.61% of patients from the total sample and 68.97% of patients from the sample with a supra-threshold life event reported a protective event belonging to the social-relational domain. We also found 51.52% and 58.62%, respectively, in the family-related domain.

Moreover, 45.45% of all patients and 51.72% of the population with supra-threshold life events either traumatic or protective reported them to be in the personal domain.

Although studies concerning the role of protective events are still scarce, research to date emphasizes the importance of protective life events of a social-relational nature. Subjective experience during adolescence is strongly characterized and influenced by the relationships with peers and their nature [18]. In addition, other strongly determining elements are the family environment and bonds. The family assumes a protective role not only directly, e.g., by avoiding critiques about body shape, but also indirectly as it helps to create self-esteem in the patient, which enables the criticism of images of excessively thin bodies from social media, the construction of her body positivity, and the regulation of emotional processes [17]. Strengthening resilience and social support have an important role in weakening and reducing the adverse effects of negative life events on adolescents and further maintaining and improving their quality of life [30,31].

The second aim of the present study was to assess the relationship between the presence of life events, either traumatic or protective, and the severity of RED and patients’ psychological maladjustment. There were no statistically significant correlations between the presence of life events in the 12 months before enrollment and the patient’s clinical severity measured according to the Eating Disorder Risk Composite and General Psychological Maladjustment Composite variables. In our data, the presence of supra-threshold life events in the Coddington Life Events Scale in the last 12 months did not correlate with the subsequent clinical worsening of the patient. This is in line with the literature in which traumatic events related to the later development of REDs are often found in the childhood period, being emotional abuse or maltreatment for the most part [32]. Indeed, according to some authors [16], the presence of childhood traumatic events is involved in the development of biological changes in the stress response system. This leads not only to a greater susceptibility to further traumatic experiences but also to a different ability to cope with stressful events, such as REDs themselves.

On the other hand, the role of protective factors in development, recovery and follow-up is still not fully clarified. One study identified the creation of a supportive environment, which promotes autonomy and satisfies the psychological demands of the patient at the same time as an incentive for patients to improve their clinical condition [33]. In the present study, questions from the CLES investigated precisely the environment surrounding the patient and how this was perceived by the patient; positive factors were, for instance, meeting people in the previous 12 months who had believed in the patient’s abilities, who had helped her to achieve her goals, or who had made her feel part of a group.

In the third aim of the study, we considered protective and traumatic events separately in relation to the severity of REDs and psychological maladjustment. A statistically significant correlation between GPMC and the presence of traumatic life events in the past year was found. This statistical evidence allows us to state that having experienced at least one traumatic event of whatever nature led to clinically elevated REDs psychological maladjustment in a higher percentage of cases than in the group of patients with no history of traumatic life events. These results are consistent with the literature stating that a life event, or the succession of several life events, causes psychological maladjustment which increases susceptibility to the development of REDs [11,12]. It is worth noting that the EDI-3 General Psychological Maladjustment Composite refers to psychological scales that provide relevant information about the psychological characteristics often present in patients with eating disorders. Therefore, it does not give a measure of maladjustment in all psychological domains but instead focuses on specific psychological features that are linked to the eating disorder itself. Our results therefore highlight the importance that traumatic life events, even when they are experienced in the recent period, can play in the dynamic development of REDs in adolescence.

No statistically significant results were obtained in the correlation between the EDRC severity parameter and the presence of traumatic life events. However, the percentages obtained in the study do suggest that more attention should be paid to the anamnestic collection of data concerning life events, especially in the family context, which may have influenced the history of the pathology.

Additionally, no statistically significant results were obtained in the correlation between the severity parameters and the presence of protective life events. Thus, having reported at least one protective event of any kind in the year preceding enrollment was not decisive for the achievement of a non-clinical outcome. This result may be strongly influenced by the limited sample of patients and by considering protective life events in the last year after enrollment. Studies concerning the role of protective events in the emergence and development of REDs are still scarce; however, it is important that clinicians seek them over a long period of time, recognize them, and can use them to promote patient outcomes. Indeed, to promote patients well-being, clinicians should reinforce their social cohesion in a peer group where physical appearance or thinness are not so important and plan with the patient achievable goals giving them the right salience. [17].

The present study contributes to the recent literature by emphasizing the close correlation between life events and the development of REDs. The peculiarity of the present work concerns the timing of life events, which to date have been studied mostly in early childhood. The literature agrees on the influence of early traumatic experiences on the subsequent development of an eating disorder; however, little attention has been paid to recent traumatic life events experienced by adolescent patients. Our study emphasizes the relevance of traumatic life events occurring in adolescence which seem to increase the patients’ psychological maladjustment.

Concerning the clinical practice with adolescent patients, our findings give an indication not to limit the investigation of traumatic life events to childhood but also to explore the recent period in order to better tailor the treatment and to focus it on the effects of exposure to the traumatic event early on in relation to the pathology. Currently, the most effective treatment for eating disorder is represented by psychotherapy, while the use of drugs has shown only limited effectiveness in the treatment of anorexia nervosa and requires close monitoring of the medical condition [34]. Regardless of the type of psychotherapy applied—for instance, in Italy, national guidelines indicate the family therapy for anorexia nervosa (FT-AN) as first choice treatment of children and adolescents [35]—it could be useful to consider the role of traumatology in the therapeutic work.

On the other hand, it could be interesting to investigate whether therapeutic approaches that aim at treating young patients exposed to traumatic life events also target RED symptoms or psychological characteristics often associated to eating disorders. A meta-analysis [36] compared the effectiveness of several psychological therapies that are used in the treatment of traumatic life events for children and adolescents. However, the studies included were focused on PTSD and associated negative emotional, behavioral, and mental health outcomes and did not consider other diagnoses or symptoms.

Furthermore, our study is innovative in that it pays attention not only to the traumatic nature of life events but also to their protective nature, which appears to play an important role in the definition of personalized treatment. Indeed, protective life events could be used as strengths in therapeutic intervention with RED patients.

Finally, the clinicians’ attention must also be focused on the patients in their care to notice if they experience traumatic life events and to intercept their needs before the disorder becomes full-blown or, if already present, worsens. Future studies may be useful in trying to decrease the time gap between the onset of the first symptoms and the diagnosis of the disorder, which could in turn lead to a better outcome of the disorder.

Regarding study limitations, first, the sample size is relatively small. In the future, the results of our study need to be replicated in larger samples of patients. Second, the CLES has certain limitations. It is a recent test with great potential but still little supporting literature. Moreover, it refers to the 12 months before enrollment, but it would also be interesting to study earlier periods, e.g., to better clarify the role of traumatic events that occurred in the childhood period, to which numerous articles in the literature refer.

To conclude, future studies could build on these preliminary data and investigate the role of life events in other psychiatric disorders as well.

## Figures and Tables

**Figure 1 children-10-00376-f001:**
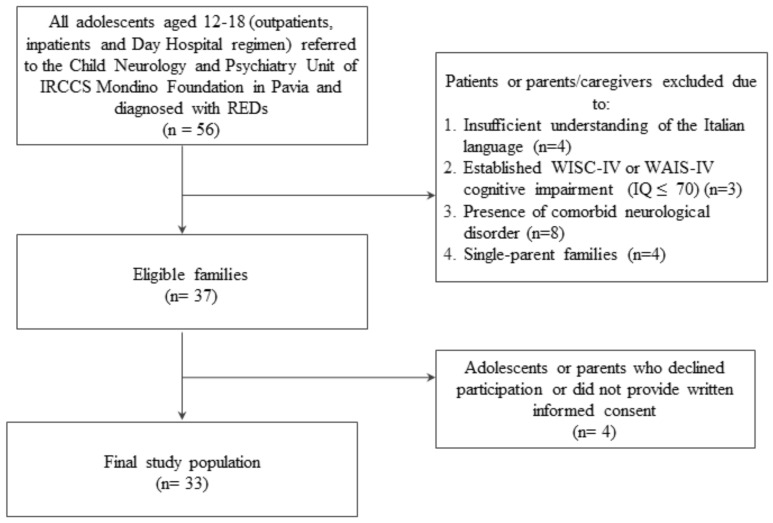
Flowchart of the study population.

**Figure 2 children-10-00376-f002:**
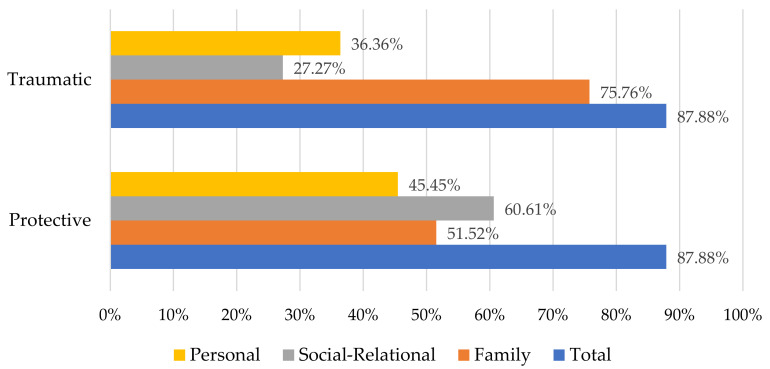
Presence of life events and their nature.

**Figure 3 children-10-00376-f003:**
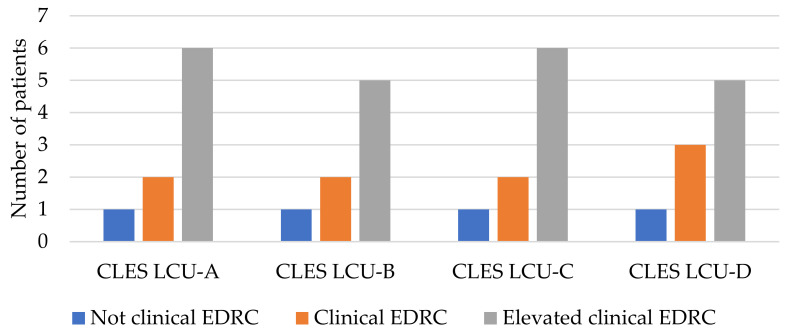
Correlation between supra-threshold CLES in each of all four periods and EDI-3 EDRC.

**Figure 4 children-10-00376-f004:**
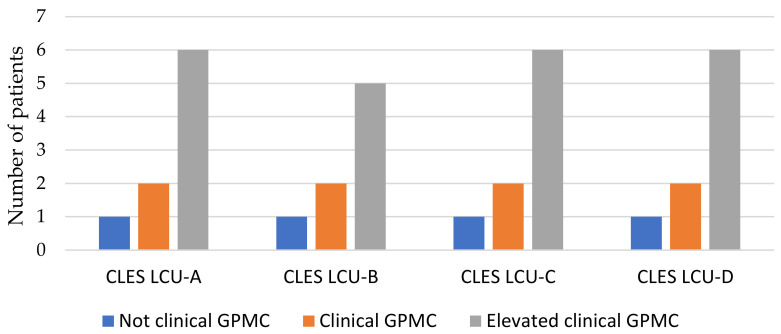
Correlation between supra-threshold CLES in each of all four periods and EDI-3 GPMC.

**Table 1 children-10-00376-t001:** Frequencies of the presence of traumatic events and EDI-3 GPMC severity.

	Not Clinical EDI-3 GPMC (*n* = 5)	Clinical EDI-3 GPMC (*n* = 6)	Elevated Clinical EDI-3 GPMC (*n* = 22)
Presence of family-related TE, *n* (%)	2 (40%)	4 (66.67%)	19 (86.36%)
Absence of family-related TE, *n* (%)	3 (60%)	2 (33.33%)	3 (13.64%)
Presence of social-relational TE, *n* (%)	0 (0%)	3 (50%)	6 (27.27%)
Absence of social-relational TE, *n* (%)	5 (100%)	3 (50%)	16 (72.73%)
Presence of personal TE, *n* (%)	2 (40%)	3 (50%)	6 (27.27%)
Absence of personal TE, *n* (%)	3 (60%)	3 (50%)	16 (72.73%)

Abbreviations. TE: traumatic events.

## Data Availability

Data are available upon reasonable request in Zenodo (10.5281/zenodo.7348803).

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
