# Peer review of "Life Events in the Etiopathogenesis and Maintenance of Restrictive Eating Disorders in Adolescence"

_children, 2023, doi:10.3390/children10020376_

Round 1

Reviewer 1 Report

The work presented for the review is original. The title of the work:  Life events in the etiopathogenesis and maintenance of Restrictive Eating Disorders in adolescence „     corresponds to its content.

The main goal of this study was to explore in a sample of adolescent patients with Restrictive Eating Disorders the presence of life events in the year before enrolment and to characterize them.

The introduction corresponds to the study issues and is pertinent to the issues discussed. The research material and research methods are sufficient. The analysis of the results and the interpretation of the results are correct.  The cited literature is correct and provides for the most recent  foreign literature items. Citations in the content of the publication are fully covered in the items included in the Bibliography and vice versa. The conclusion and the  recapitulation are appropriate. An important limitation of the research is the relatively small sample. The work can be accepted after considering minor editorial corrections.

Author Response

Response to Reviewer 1 Comments

Point 1: The introduction corresponds to the study issues and is pertinent to the issues discussed. The research material and research methods are sufficient. The analysis of the results and the interpretation of the results are correct.  The cited literature is correct and provides for the most recent  foreign literature items. Citations in the content of the publication are fully covered in the items included in the Bibliography and vice versa. The conclusion and the  recapitulation are appropriate. An important limitation of the research is the relatively small sample. The work can be accepted after considering minor editorial corrections.

Response 1: Thank you very much for taking the time to read the manuscript. We appreciate the suggestions that helped make the manuscript better.

Reviewer 2 Report

The paper raises a good research question, a sound method is used and the results are presented in a clear manner. The study limitations are pointed, as well as future lines of research.

Author Response

Response to Reviewer 2 Comments

Point 1: The paper raises a good research question, a sound method is used and the results are presented in a clear manner. The study limitations are pointed, as well as future lines of research.

Response 1: The authors thank Reviewer 2 for taking the time to read the manuscript and for his/her interest.

Reviewer 3 Report

The article is qualified and comprehensive. however, I suggest it be repeated with a study in which the number of samples is increased. The article can be published as it is. congratulations

Author Response

Response to Reviewer 3 Comments

Point 1: The article is qualified and comprehensive. however, I suggest it be repeated with a study in which the number of samples is increased. 

Response 1: The authors thank Reviewer 3 for his/her interest and the suggestion regarding future studies, which would be interesting, and the research group is moving in that direction.

Reviewer 4 Report

I congratulate the author(s) for the work done. I think it is an interesting and necessary work. I believe it should be published and is relevant to the scientific community. However, for its better visibility and readability, I make constructive recommendations, in the best academic spirit.
The title is suggestive and appealing, and academically, it contains perfectly the summary of what the reader will find in the text.
The summary is good and attractive, and contains all the important sections of the scheme Introduction-Objectives-Methodology-Results-Conclusions. However, I think it would be very useful and suggestive to introduce some more methodology, especially to see why this text is novel or contributes to the academic community and why it can be extrapolated from your country to the rest of the world.
The key words are very well chosen.
A harmonious distribution of paragraphs should be sought, so that they are all of a similar length, 6-7 lines, with no very long paragraphs. This will make the text more readable and understandable even if it is already well written. For example, the first paragraph of point 1, for example, is so short that it seems to be loose, orphaned.
In the Introduction and in the Conclusions, it would be necessary to defend more clearly and directly why a useful methodology is being offered to the scientific community, why it is innovative or how a methodology that already existed has been updated (as we read in Methods), and how this methodology can be replicated by other researchers. This does not appear and with all that is written on this topic, it is necessary to differentiate oneself and this text can do so.
The works cited as a theoretical framework or literature review are very few and more context would be lacking, precisely to differentiate themselves. Also, although they are solid and prestigious, some are very old for the area and there are few recent and international citations for a journal of this prestige and for this topic. I recommend adding 5-6 super up-to-date references, from top-level international journals, especially when an article on this subject is being offered, and that these texts are from 2022 only.
The Results section is very good, as the exposition and argumentation are very well exposed and spun. It is the best part of the text and I congratulate you on the exposition.
It lacks a section on limitations and foresight. It should be a separate section, after the conclusions and discussion. This information is relevant and enriches any good article, even more so in a publication of this prestige.
The tables and figures are in a different font, different from that of the layout and the text. Perhaps it would be a good idea to review the journal's rules to see if they could be more in line with the rest of the typefaces.

Author Response

Response to Reviewer 4 Comments

Point 1: The title is suggestive and appealing, and academically, it contains perfectly the summary of what the reader will find in the text.

Response 1: Thank you for your interest and for taking the time to read the manuscript. We appreciated the points you raised.

Point 2: The summary is good and attractive, and contains all the important sections of the scheme Introduction-Objectives-Methodology-Results-Conclusions. However, I think it would be very useful and suggestive to introduce some more methodology, especially to see why this text is novel or contributes to the academic community and why it can be extrapolated from your country to the rest of the world. The key words are very well chosen.

Response 2: We thank the reviewer and enriched the abstract as suggested, trying to stay within the required number of words

Point 3: A harmonious distribution of paragraphs should be sought, so that they are all of a similar length, 6-7 lines, with no very long paragraphs. This will make the text more readable and understandable even if it is already well written. For example, the first paragraph of point 1, for example, is so short that it seems to be loose, orphaned.

Response 3: We agreed with the reviewer and redistributed the paragraphs.

Point 4: In the Introduction and in the Conclusions, it would be necessary to defend more clearly and directly why a useful methodology is being offered to the scientific community, why it is innovative or how a methodology that already existed has been updated (as we read in Methods), and how this methodology can be replicated by other researchers. This does not appear and with all that is written on this topic, it is necessary to differentiate oneself and this text can do so.

Response 4: Reviewer 4 raised an interesting point. We highlighted the innovativeness of the study and methodology in the Introduction and Discussion sections.

Point 5: The works cited as a theoretical framework or literature review are very few and more context would be lacking, precisely to differentiate themselves. Also, although they are solid and prestigious, some are very old for the area and there are few recent and international citations for a journal of this prestige and for this topic. I recommend adding 5-6 super up-to-date references, from top-level international journals, especially when an article on this subject is being offered, and that these texts are from 2022 only.

Response 5: We agreed with the reviewer and take the opportunity to deepen the literature. We added more recent works in Introduction and in Conclusion sections (2,6,7,8,11,17,27,28).

Point 6: The Results section is very good, as the exposition and argumentation are very well exposed and spun. It is the best part of the text and I congratulate you on the exposition.
It lacks a section on limitations and foresight. It should be a separate section, after the conclusions and discussion. This information is relevant and enriches any good article, even more so in a publication of this prestige.

Response 6: We agreed with the reviewer, and we highlighted limitations and future directions in a separate paragraph at the end of the Discussion section.

Point 7: The tables and figures are in a different font, different from that of the layout and the text. Perhaps it would be a good idea to review the journal's rules to see if they could be more in line with the rest of the typefaces.

Response 7: We thank the reviewer for raising this point. We corrected the layout through all the tables and figures.

Reviewer 5 Report

Dear authors,

Thanks for the study, the idea of assessing the relationship between life events and restrictive eating disorders is interesting and relevant, but I think this study needs some refinement:

1. Add values to Figure 2 to avoid duplication of information (number and percentage) in the text.

2. The information in Figure 3 is not clear. I do not understand what the author is trying to show/say. I would suggest to think of another way to present the existing information.

3. What do these terms mean: clinical, not clinical, elevated clinical in Figures 4 and 5, and Table 1. No information is provided in the methodology.

4. The time periods to be assessed are 0-3 months, 0-6 months and 0-9 months and 0-12 months. Therefore, it is not clear why, for example, Figure 4 showed a lower value (number of patients) for the period 0-12 months than for the period 0-9 months, since it included this period.

5. Abbreviations are used extensively, making the text difficult to understand. Perhaps the text could be revised and some corrections made.

6. Use a consistent style in the publication and provide numerical information in the form of figures (L 136-137), as in the previous text (L 127-133).

7. Shouldn't the self-report questionnaire be mentioned as a limitation because it is the patient's interpretation?

Author Response

Response to Reviewer 5 Comments

Point 1: Add values to Figure 2 to avoid duplication of information (number and percentage) in the text.

Response 1: We thank the reviewer for taking the time to read the manuscript. We corrected the Figure according to Reviewer’s suggestion.

Point 2: The information in Figure 3 is not clear. I do not understand what the author is trying to show/say. I would suggest to think of another way to present the existing information.

Response 2: We agreed. The graphic was confusing, we preferred to delete it and report the data in the text avoiding redundancies.

Point 3: What do these terms mean: clinical, not clinical, elevated clinical in Figures 4 and 5, and Table 1. No information is provided in the methodology.

Response 3: We clarified the meaning of non-clinical, clinical, and high clinical in the Procedures and Methods section.

The outcome is non-clinical when the result obtained in each composite scale gives a score below the 70th percentile, clinical when the score is between the 70th and 85th percentile and high clinical when is above the 85th percentile”.

Point 4: The time periods to be assessed are 0-3 months, 0-6 months and 0-9 months and 0-12 months. Therefore, it is not clear why, for example, Figure 4 showed a lower value (number of patients) for the period 0-12 months than for the period 0-9 months, since it included this period.

Response 4: Thank you for your suggestion. We better explained periods’ cut-offs.

Each of these four periods has a specific cut-off. The score awarded is full when the event occurred in the previous 3 months, ¾ of the total when it occurred 4-6 months earlier, 1/2 if it occurred in the previous 7-9 months and, finally, ¼ if it occurred between the previous 10 and 12 months. Scores resulted positive, and patients were considered at risk of developing physical or emotional disorders (suprathreshold CLES) if they exceed the LCU-appropriate and the age-appropriate 75th percentile”

Point 5: Abbreviations are used extensively, making the text difficult to understand. Perhaps the text could be revised and some corrections made.

Response 5: We followed the suggestion and made the text more fluent.

Point 6: Use a consistent style in the publication and provide numerical information in the form of figures (L 136-137), as in the previous text (L 127-133).

Response 6: We added numerical information in Figure 2 to make the results section clearer and avoid redundancies.

Point 7: Shouldn't the self-report questionnaire be mentioned as a limitation because it is the patient's interpretation?

Response 7: Since the focus of the self-report questionnaire is the subjective perception of life events, either positive or negative, as having an impact on the person, we considered that in this study the self-completed questionnaire was a truthful representation of the patient's experience.

Round 2

Reviewer 5 Report

Dear authors,

Thank you for your improvement of publication. The publication has been published in journal Children.

Good luck with your future research!

Author Response

Thank you